# How Does Toddlers’ Engagement in Literacy Activities Influence Their Language Abilities?

**DOI:** 10.3390/ijerph19010526

**Published:** 2022-01-04

**Authors:** Raffaele Dicataldo, Maja Roch

**Affiliations:** Department of Development and Socialization Psychology, University of Padua, 35131 Padua, Italy; maja.roch@unipd.it

**Keywords:** home literacy environment, engagement, vocabulary, morphosyntactic skills, toddlers

## Abstract

The most intensive period of language development is during the first years of life, during which the brain is developing rapidly. Research has shown that children from disadvantaged households who received high-quality stimulation at a young age grew into adults who earned an average of 25% more than those who did not receive these interventions. In addition, it has been suggested that children who show a greater interest in literacy-related activities and voluntarily engage in them are likely to become better readers than children with less interest in literacy. These children’s factors, along with their engagement in literacy activities, are important components in children’s early literacy experiences and may affect their early language development. In this study, we examined associations among maternal education, home literacy environment (HLE), children’s interest and engagement in literacy activities, and language development of 44 toddlers aged between 20 and 36 months. Overall, results showed that only children’s engagement in literacy activities was related to vocabulary and morphosyntactic skills, whereas maternal education, HLE, and children’s interests were not. These results suggest that taking advantage of individual children’s interests by planning activities in which children are fully engaged, may be effective strategies for promoting children’s oral language development.

## 1. Introduction

Speech and language are the most important skills that we use to communicate with others. Infants are rapid learners of language and develop these skills during the first years of life [1]. Children vary in their development of speech and language; however, there is a natural progression or “timetable” for mastery of these skills for each language. The most intensive period of speech and language development is during the first three years of life, a period when the brain is developing and maturing rapidly [2]. These skills appear to develop best in a world that is rich in sounds, sights, and consistent exposure to the speech and language of others [3].

The evolution of theoretical models of language acquisition has led to the progressive transition from defining language development as an intra-individual and intrapsychic phenomenon, as supported by the Innatism [4], to an inter-individual and interactive one, as supported for instance by Vygotskij’s theory and Neuroconstructivism [5,6]. Today, it is widely accepted that children acquire language during interactions with others in significant contexts in which they grow up [7]. According to Neuroconstructivism, developmental processes (including those related to language development) gradually emerge from complex cascades of biological and physical interactions [6]. One way of conceptualizing this is to view development as experience-driven processes that occur within complex biological and ecological systems—and thus as constrained by internal and external factors at multiple levels (genetic, neural, behavioral, societal) and timescales [8]. Finally, Neuroconstructivists argue that development is an experience-driven process, and that developmental systems are adaptive systems that change in response to their environments. Coherently to this view, language skills develop through learning experiences.

Numerous studies show that linguistic input, in terms of its quantitative and qualitative properties, is critical to various aspects of development, including lexical, syntactic [9,10], and social development [11]. Additionally, literature has been interested in the influence of the family environment on child language development, highlighting some variables mostly related to atypical developmental trajectories. This research focuses on the importance of specific home characteristics and parent-child book reading activities for preschool children’s language and literacy development showing a positive link between home characteristics, parents’ involvement in home literacy practices, and children’s later language and literacy skills [12,13].

However, according to a Bioecological theoretical framework, not only environmental variables contribute to language development but also personal characteristics, such as a child’s interest in books and involvement in literacy activities [14]. Few studies have adopted this framework for deeply exploring the role of some children’s characteristics in relation to their language development. Our aim in this work is to integrate these different aspects to reach a more complete view of the role of maternal education, home literacy environment and children’s interest and engagement in literacy activities in toddlers’ vocabulary and morphosyntactic silks, and in comprehension and production, evaluated through direct and indirect measures.

### 1.1. The Role of Maternal Education on Early Language Development

Family socioeconomic status (hereafter, SES)—generally measured by parents’ education, occupation prestige, and household income or a combination of these factors—refers to a measure of the economic and social position of a family in relation to the level reached by the Society in which they live. SES is predictive of a broad range of important life outcomes as intelligence, academic achievement, and school readiness of kindergarteners [15]. Although SES indicators are highly interrelated, specific components of SES may influence child outcomes through specific mechanisms [16]. Empirical evidence suggests that parental education, in particular maternal education, has a unique causal influence on children’s language development and later academic outcomes [17]. Higher levels of maternal education are positively associated with different academic outcomes for children throughout development. Prior to children’s school entry, higher maternal education has been associated with more advanced spontaneous language production [18] and with higher performance on standardized cognitive achievement tests [19].

Although this research highlights a positive association between higher levels of maternal education and children’s academic outcomes across childhood and adolescence, it does not necessarily imply that maternal education is the cause of children’s outcomes. Maternal education, in fact, is associated with different characteristics—including family background, genetics, and availability of material resources—in turn, associated with child outcomes. Maternal education is a purely descriptive environmental predictor of language and reading development, whereas storybook exposure constitutes a plausible mechanism by which differences in home education level operate on development. Not surprisingly, maternal education and maternal beliefs about the importance of literacy experiences for children have been associated with more frequent book reading activities, having positive effects on child language [20,21].

Overall, the evidence reported above suggests the importance of parental education, in particular maternal education, and demonstrates the need for further empirical evidence exploring the key mechanisms of the relationship between maternal education and children’s linguistic outcomes [22].

### 1.2. The Role of Home Literacy Environment

The experiences, attitudes, and materials related to the first literacy that a child encounters and interacts with at home, refer to what is called a “home literacy environment” [23]. The HLE usually refers to activities undertaken by family members at home related to literacy learning [24,25,26,27] as well as literacy resources at home and parental attitudes toward literacy activities [28,29].

A considerable number of studies have investigated the relationship between HLE and children’s language skills [30], examining how specific aspects of the environment can influence the acquisition of children’s communicative and linguistic skills and, subsequently, the development of literacy. Previous studies found that, for instance, the availability of educational materials at home supports children’s language and literacy skills [31]. The number of pictured books at home predicts children’s receptive language skills and expressive vocabulary [26]; familiarity with books correlates with the later vocabulary of pre-schoolers as well as early reading skills [32]. Presumably, easy access to children’s books facilitates shared book reading activities [21].

HLE refers also to the frequency of story reading activities [33], letter naming activities [34], parental support during the activities, and beliefs for such activities [35]. It is usually measured by rating scales and can be divided into “formal” and “informal” home-based literacy interactions [30]. HLE and parent-child involvement in home literacy activities are essential for the growth of children’s language and emerging literacy. A rich and supportive HLE allows more rapid and accurate acquisition of new words, a greater phonological awareness, as well as a knowledge of alphabetical symbols and their organization within the text during reading. Book reading activities are particularly beneficial for the development of linguistic and narrative skills [32]. These literacy-related practices represent also strong emotional and cultural experience since they offer children the opportunity to look out over a fantastic and imaginary world, experiencing moods of characters and exercising the ability to predict events in a process of socio-cultural construction capable of conveying values and ideals of the individual and the community [36].

The measure of the frequency of shared book reading represents the more powerful predictor of children’s literacy achievement [12]. Research indicates that shared-book reading supports a range of early language skills including phonological awareness [37,38], vocabulary [39], syntactic development [40], narrative skills [41], print awareness [42], and reading ability [12]. In addition, Sonnenschein and Munsterman [43] found that whereas reading frequency correlated with phonemic awareness and print orientation, the affective quality of reading interactions predicted children’s motivation for reading.

All these findings point out that HLE plays a crucial role in determining the child’s degree of cognitive and language development, and later school readiness and success [36]. Moreover, it seems obvious, that a stimulating HLE especially in the first phases may produce better developmental outcomes and prevent or reduce the impact of other environmental factors.

### 1.3. The Role of Children’s Interest and Engagement in Literacy Activities

The Bioecological framework suggests that children’s development is not determined exclusively by the characteristics of the surrounding environment [14]. It is important to consider the active role that a child plays in choosing their own experiences, from an early stage of development [44]. Child characteristics, in fact, are important for language development, including physical and mental health, socio-emotional skills, and approaches to learning [45]. In this work we were interested in child interest and engagement in literacy activities since, as argued by Whitehurst and Lonigan [46], “a child who is interested in literacy is more likely to facilitate shared-reading interactions, notice print in the environment, and spend more time reading once he or she is able” (p. 854), with effects on language development.

Opportunities to engage in meaningful interactions with interesting and appealing materials and literacy-related activities are critical to successful literacy achievement [47]. Children dedicate more attention and effort to activities in which they are interested [48]. Children’s early interest in reading is thought to be critically important to their later literacy development and school success [49]. Scarborough and Dobrich [13] suggest that children who are interested in reading may engage in more reading and literacy-related activities and thus may become better readers than similar children with less interest. Additionally, children who are interested in reading influence parents to read to them more frequently or for longer [50]. There is growing evidence that literacy interest plays an important role in early literacy skills [51] suggesting that children’s literacy interest may be important to any model that links shared reading with language development and literacy achievement [49].

An additional children characteristic that seems related to their linguistic outcomes is children’s engagement or involvement during storybook reading activities. A study of precocious talkers found that child involvement correlated with measures of oral language at age 2 and print concepts at age 4 [33]. Child engagement was a better predictor of literacy than either the type or frequency of parental behaviours during the reading activities. They concluded that child engagement could not be explained by intelligence or by exposure to reading, suggesting that engagement may be an independent factor that plays a role in later literacy [33].

Children’s engagement seems to play an important role in the quality of picture book reading. Young children (ages 9, 17, and 27-months) with high levels of vocalizations were asked more questions and provided with more feedback from parents than children with lower rates of vocalizations [52]. Certainly, individual differences in children’s engagement in literacy are likely to affect the quality of shared-book reading [13,50], yet only a few studies have examined young children’s involvement in literacy [53,54].

Children’s contribution to shared reading processes has not been extensively explored, consequently its role is not as well understood and thus, the determinants of interest and engagement in home literacy activities have been grossly under-researched. The few studies that have investigated children’s interest and engagement in literacy activities indicate that are associated with language development [55], specific emergent literacy knowledge such as letter knowledge [24] and vocabulary [56], and early reading status [38]. Results from a recent meta-analysis indicate that variations in young children’s interest are related to differences in almost every literacy and language outcomes [57].

Additionally, since children’s attitudes towards reading during primary school are associated with their willingness to select challenging reading material and their academic standing [58,59], highlighting relative stability over time, children interest and engagement during literacy activities should be considered, as two separate constructs, in any model that links shared reading with later literacy achievement.

### 1.4. Relations among SES, HLE, Child Interest and Engagement in Literacy Activities

As can be easily understood, SES, HLE, child interest, and engagement in literacy activities do not act in isolation but interact with each other to form a complex interweaving that determines the developmental environment of each child.

An important mechanism by which SES operates on children’s language development is the literacy-related resources and interactions that young children experience at home, especially during the preschool years [12]. Parents, in particular mothers, are the first teachers of their children [60,61] and they are uniquely motivated to promote their children’s wellbeing and development [62]. Family SES may be considered as one of those static factors that influence the features of the HLE for pre-schoolers. As previously reported, maternal education and family income are associated with variations in the environment and experiences of children, including the quantity of language that they hear [63]. In detail, family income affects the availability of material resources that can influence the quantity of input and thereby child development, whereas parental educational history is linked to non-material resources, including the parent’s academic competence, attitudes toward education [64], knowledge and beliefs about child development [65], and overt behaviour [66]. Although less frequent reading generally has been reported in low-income families, wide variation in literacy-related activities has been documented in these households [67]. The literature reported above shows that low SES seems to be a risk factor for HLE and therefore early literacy skills of children, however, Payne and colleagues [26] state that there are also variations in the quality of HLE among low SES families. The researchers stress the fact that, despite many economic burdens, many low SES families put effort into literacy support through, for example, shared book reading [68]. To date, there is a considerable body of evidence pointing to the importance of HLE in supporting young children’s development of early literacy skills [69], arguing that some aspects of HLE may be even more important than family SES factors in predicting children’s outcomes [12]. Measures of the frequency of shared book reading represent a more powerful predictor of children’s literacy achievement than measures of family SES [70].

Concerning children’s engagement in literacy activities, as reported below, it seems to play an important role in the quality of picture book reading affecting the frequency and quality of shared book reading. These individual differences in children’s engagement in literacy are likely to affect early literacy skill development [51], particularly vocabulary and grammar development [57]. Previous studies suggest that children’s interest and engagement in literacy-related activities are relevant predictors of language development but do not address the relationship of child interest to HLE, though it is not clear how children’s interest promotes learning.

There is growing evidence that literacy interest and engagement play an important role in early literacy skills [51] and that child literacy interest is associated with the HLE [71], although the direction of the association is yet to be established. One possibility is that children’s interest facilitates a more active engagement in the reading process and that, in turn, promotes language use, thus children’s interest during literacy activities mediates the relationship between HLE, operationalized as parental involvement in literacy activities, and linguistic outcomes in pre-schoolers [56]. A second possibility is that HLE contributes to children’s interest in literacy activities [24,29,56]. Çakmak and Yılmaz [72] investigated the relationship between children’s interest in books and HLE among 50 middle and high-SES kindergarteners and found that rich HLEs resulted in increased interest in print materials and reading behaviours. Scarborough and Dobrich (1994) found that pre-schoolers’ perceived interest in literacy had a stronger relationship to children’s language and literacy outcomes than did measures of the frequency and quality of shared book reading activities. Additionally, there are results reporting that children’s interest and HLE are independent factors [73,74,75]. Therefore, to date, whereas the relations between SES and HLE are clearer, the literature provides mixed results concerning the relations between HLE, children’s interest and engagement in literacy-related activities, and linguistic outcomes. Less research has focused on the ways in which individual characteristics, specifically children’s interest and engagement in literacy activities, are related to language development. Subsequently, it seems necessary to analyse further the links between environmental factors and children characteristics producing effects on language outcomes, particularly in the first stages of language development.

### 1.5. The Current Study

The aim of this study is to extend previous findings of the associations among aspects of the home literacy environment, children’s interest and engagement in literacy-related activities, and oral language skills, namely vocabulary and morphosyntax, of toddlers aged between 20 and 36 months. We expanded upon previous research by: (a) targeting the earliest stages of the influence of family reading behaviour by focusing on the pre-kindergarten period; (b) using shared-reading frequency as a measure of family reading behaviour; (c) using a parent-report measure of child interest and engagement in literacy activities; (d) investigating the relationship between maternal education, family reading behaviour, children interest and engagement in literacy activities—as two separate constructs—vocabulary and morphosyntactic skills in production and comprehension. Our aim is to clarify the relationships of these variables in a bioecological framework and quantify the weight that each variable has on the vocabulary and morphosyntactic skills of toddlers.

There were two specific research questions in this study:

RQ1. The first research question was “What relationships exist between maternal education, frequency of literacy-related activities, books’ availability, children’s interest and engagement in literacy activities?”

We expected, in line with previous studies (Reese et al., 2010), that maternal education would be related to books’ availability and to the amount of shared book reading. Low relations were expected between family reading behaviours, children’s interest, and engagement in literacy-related activities, considering these as individual characteristics independent from SES and HLE, thus related more to children’s predisposition to learn.

RQ2. The second research question concerned two points. The first point was “What are the relationships among maternal education, family reading behaviours, children interest and engagement in literacy-related activities, children’s age, vocabulary and syntactic comprehension (RQ2a)?”. The second point was “Are the relationships among these variables similar concerning language production (RQ2b)?”

Overall, we expected that both context variables (e.g., maternal education and literacy experiences at home) and personal characteristics (e.g., age, child interest for literacy and engagement) would contribute to children’s early oral language skills (e.g., vocabulary and morphosyntactic skills, both receptive and expressive). In detail, we expected that children’s personal characteristics would be more related to the oral language skills in production. Children who are interested in reading may engage in more reading and literacy-related activities. Moreover, greater involvement of the child in literacy activities provides for greater interaction between parent and child and therefore more opportunities for communicative exchanges, feedback, and adjustments that can promote children’s language production.

## 2. Materials and Methods

### 2.1. Participants

Forty-eight toddlers aged between 18 and 36-months and their parents were involved in this study. The exclusion criteria were preterm birth, birth complications, visual/auditory impairment, neurodevelopmental disorders, and medical problems. Two children were dropped since they did not complete the evaluation of oral language skills. Two parents refused to return the questionnaires used to assess SES and HLE, so they were not included in the following analyses. The final group included forty-four 20-36-months-old children (Mage = 28.2 months, SD = 5.2, boys *n* = 28) and their mothers. According to parent reports, none of the children had cognitive impairments or language difficulties and none of them had ever been referred to the National Health Services. The Ethics Committee of the University of Padua approved this study (protocol n. C4CA28B3AA0D117A819B839BEE8E7181).

### 2.2. Procedure

We recruited children in four pre-kindergartens at the beginning of the school year. Before the assessment of children’s oral language skills, the investigator spent several hours in schools to interact with all the children and establish an initial contact necessary to obtain their collaboration during the test sessions. The investigator joined the educators in all the activities carried out during the morning: welcoming, breakfast, free-play, and shared-book reading. Children were assessed individually in a quiet room during the school days in the presence of an educator. Additionally, mothers completed a questionnaire focused on family characteristics, HLE, and children’s interest and involvement in literacy-related activities, and filled a standardized test used to assess children’s early oral language skills.

### 2.3. Measures

#### 2.3.1. Family SES, HLE, Child Interest, and Engagement in Literacy Activities

Information about family socioeconomic status, home literacy environment, and child interest in literacy-related activities were collected through the SES/HLE questionnaire, developed by Roch, Florit, & Levorato (unpublished), consisting of 30 items. The first section includes items related to child characteristics, namely age and gender, and socioeconomic indicators such as parental education level and annual income. For the aim of this study, as reported above, we used maternal education measured in years, as an indicator of SES. The subsequent items are related to HLE, child interest, and engagement in literacy activities. We asked parents to report the number of children’s books at home rated on a Likert scale ranging from 0 to 5 where 0 means “none” and 5 “more than eighty”. Additionally, we asked parents to report the frequency of the most carried out literacy activities at home, namely reading books, use of educational games, and telling stories, rated on a scale ranging from 0 to 5 points where 0 means “never” and 5 “more than once a day”. We used the number of books for children at home and the frequency of shared book reading activities as indicators of HLE. Finally, we asked parents to report the frequency with which their child engages with books independently and the frequency with which their child, during reading, dialogues with the adult, interrupts and invents his stories. Scores in these four items were rated on a scale ranging from 0 to 3 where 0 means never “never” and 3 “a lot”. The score obtained at the first item was used as a measure of child interest in reading whereas scores obtained through the scoring of the later 3 items were used to obtain a measure of child involvement during literacy activities.

#### 2.3.2. Children’s Oral Language Skills

Children’s oral language skills were assessed, to obtain both direct and indirect measures, through two standardized tests:

Test del Primo Linguaggio (TPL) is a standardized test for Italian speakers used to directly assess a child’s communication and language skills of children between 12 and 36 months [76]. The purpose of this test is to provide a description of the main language skills emerging in the early years of life and allow the assessment of the individual child with respect to its normative group. It consists of three different scales that evaluate three aspects of language—pragmatic, semantic, and first syntax—allowing the assessment of communication and verbal skills, both in production and comprehension, in a strict sense. In this study we used two scales, namely Vocabulary Scale and First Syntax Scale. The Vocabulary Scale assesses the ability to understand and produce names based on simple figures. The material used for both the comprehension and the production tests consisted of twenty figures, representing objects of daily life, belonging to natural categories. In the comprehension trial, children had to indicate which out of four pictures best represented target words, whereas in the production one had to name figures in succession (range 0–20). The First Syntax Scale included a comprehension test of action verbs. In this trial, the child had to indicate which out of four pictures best represented target objects according to the definition of use (range 0–20). The production trail involved the presentation of twenty-two elementary vignettes depicting a child or adult performing simple, everyday actions. The child was asked to describe what he/she saw, and the average length of the enunciated product gave the score for each item (range 0–60). Scores, both in comprehension and in production, were then compared with summary tables of percentile values.

The Child’s first vocabulary (PVB) is the Italian version of the MacArthur–Bates Communicative Development Inventory (MB-CDI) [77,78]. It consists of a list of words, gestures, and sentences derived from child language samples from each language for which they were developed. It is a parent report instrument used to capture important information about children’s development abilities in early language, including vocabulary comprehension, production, gestures, and grammar. In this study the format Words and Sentences was used. It included word production (approximately 680 words) and several sections of early grammar, different for each language, that included specific morphological and/or syntactic forms (such as regular/irregular inflections or case markings) and sentence complexity. Mean length of utterance was also studied. Parents were also asked to write down the three longest utterances their child produced. Scores obtained were then compared with appropriate summary tables of percentile values, graphs, or both. Moreover, this may be calculated by hand the Age of lexical development (ALD) and the Lexical Quotient (LQ).

### 2.4. Analysis Plan

First, preliminary analyses (i.e., identification of outliers), and descriptive statistics (i.e., means, standard deviations, skewness, kurtosis, and range) were computed. Principal Component Analysis (PCA) was used to reduce the number of variables used to assess child engagement in reading activities using the scores obtained at the three items of the questionnaire SES/HLE focused on it, described above. Second, correlations were run to analyze the overall pattern of relations between variables of interest, namely maternal education, number of books available at home, frequency of shared book reading at home, child interest, and engagement in literacy activities. Correlations allowed addressing the RQ1. Additionally, a structural equation model was fitted to analyze all the possible relationships between environmental factors and children’s characteristics.

Structural equation models were used to address RQ2, i.e., to address the contribution of maternal education, frequency of shared book reading, child interest, and engagement on vocabulary and syntactic skill. In Model 2a we used a single measure of vocabulary and morphosyntactic skills in comprehension measured with TPL, therefore, observed variables were used in this model. Subsequently, Confirmatory Factory Analysis (CFA) was used to create a latent variable for expressive vocabulary including measures of vocabulary production measured with TPL and total words measured with PVB (hereafter, Expressive Vocabulary Latent variable), and for syntactic skill in production including measures of syntactic skill measured with TPL, MLU and syntactic complexity measured with PVB (hereafter, Syntactic skill Latent variable). In Model 2b we used fitted values of Expressive vocabulary Latent variable and Morphosyntactic skill Latent variable. Direct and indirect relations between variables were tested controlling for children’s age.

Model fits were evaluated by using the following multiple indices: chi-square statistics, comparative fit index (CFI), root mean square error of approximation (RMSEA), and standardized root mean square residuals (SRMR). Typically, RMSEA values below 0.08, and CFI values equal to or greater than 0.95, and SRMR equal to or less than 0.10 indicate an acceptable model fit [79]. Structural equation models were conducted with R package lavaan, version 0.4–11 [80].

## 3. Results

### 3.1. Preliminary Analyses and Descriptive Statistics

In Table 1 are reported descriptive statistics (range, means, standard deviations, skewness, and kurtosis) of children’s age and scores obtained in the two tasks used to assess children’s language. There were no extreme outliers and values of skewness and kurtosis were all within acceptable limits (Kurtosis = 3.20), except for scores at the TPL vocabulary scale [81].

Concerning TPL scores (direct assessment), children on average produced 17 words of the 20 whereas they were able to recognize on average 13 words. As for morphosyntactic skill, children correctly answered about 13 items out of 20 on the comprehension task whereas children for most of the items used one single word to describe the pictures (mean = 20) showing, however, great variability in performance in this task, as predicted in this age range.

Concerning scores on PVB (indirect assessment), parents reported that children produced an average of 375 words whereas the mean length of utterance was six words. According to the norms of this test [77], 91% of participants had average or good levels of vocabulary. The others (*n* = 6) performed below the 10th percentile but were not removed from the analyses because none of the children had cognitive impairments or were at risk for language difficulties. Children have shown great variability in the syntax complexity score, with an average performance of 27. Measures of Age of lexical development and Lexical Quotient indicated that children have a level of language development in line with the normative sample for their age.

The correlation analysis between scores obtained in these two tasks, namely a direct and an indirect assessment of children’s early oral language skills, showed correlations ranging from moderate to strong (0.53 < r < 0.81) proving a great convergent validity of these two tasks.

Table 2 shows descriptive statistics (range, means, standard deviations, skewness, and kurtosis) of contextual measures namely maternal education and two measures of the HLE. Extreme outliers were not detected, and values of skewness and kurtosis were all within acceptable limits, except for the value of kurtosis for maternal education and frequency of parent-to-child storytelling. This finding is explained by the fact that most of the mothers declared that they had achieved a master’s degree and that they tell stories to their children at least once a day.

Concerning maternal education, rated in years of education, 6 mothers achieved a high school degree, 8 a bachelor’s degree, 24 a master’s degree and 6 a higher degree such as PhD or specialization. HLE measures show that, on average, between 41 and 60 children’s books are available at home and that parents engage in shared reading activities, on average, at least once a day. These results highlight that, in this group, home environments are rich and supportive with parents fully involved in literacy-related activities.

Table 3 shows descriptive statistics (range, means, standard deviations, skewness, and kurtosis) of scores on the four-child interest and involvement-related items. Extreme outliers were not detected, and values of skewness and kurtosis were all within acceptable limits.

Concerning child interest, parents report that on average, their child engages with books independently, at least once a day. Concerning items used to assess children’s involvement in reading activities that show great variability, we run a Principal Component Analysis (PCA) to simplify our data and reduce the number of variables. Using the dodgy eigenvalue >1 approach, the analysis extracted one factor from 3 items namely “Child interrupts during reading”, “Child invents stories during reading” and “Child dialogues during reading (Factor loading: 0.689; 0.835; 791) explaining 61% of the total variance of our variable called “Child involvement”. Standardized factor scores for each child were used in the following analyses.

### 3.2. Pattern of Relations between Contextual Variables and Child Interest and Engagement

RQ1 was addressed by running a correlational analysis among the measures of contextual factors (maternal education, HLE) and child characteristics (age, interest, and engagement). Results are reported in Table 4.

Child age is moderately and positively correlated with child engagement in reading (r = 0.64) whereas is negatively correlated with child interest in reading independently (r = −0.39) showing that growing up children tend to prefer the involvement in shared book reading activities instead of engaging with books independently. Maternal education showed significant correlations with the measure of the number of books for children (r = 0.34) and with the frequency of shared-book reading (r = 0.73), which, in turn, is weakly related to the number of books at home (r = 0.33). Child interest in reading and their engagement in reading activities were not correlated with other contextual factors. These results addressed RQ1 by showing that maternal education, is related to the number of children’s books at home and to the frequency per week of shared book reading activities whereas child interest and engagement in reading are not correlated with other contextual factors.

### 3.3. Pattern of Relations between Contextual Variables, Child Characteristics, and Language Development

Three Structural equations models were tested to address the research questions. A first structural equation model was fitted to analyze the relationships that exist between maternal education, frequency of literacy-related activities, books’ availability, children’s interest, and engagement in literacy activities, i.e., RQ1 (see Model 1). Subsequently, two separate structural equation models addressing the relative contribution of maternal education, number of children’s books, frequency of reading behaviors, and children’s interest and engagement in reading activities to vocabulary and morphosyntactic skills in comprehension (Model 2a) and in production (Model 2b) were fitted to answer to RQ2a/b. In these two models, we included only significant paths of Model 1 whereas non-significant paths were not included.

Model 1 shows that maternal education was positively related to the frequency of shared book reading activities at home (ß = 0.31, *p* < 0.001, 95% CI [0.20–0.40]), accounting for 55% of the variance in the frequency of shared book reading activities at home, and to the number of books for children at home (ß = 0.14, *p* < 0.01, 95% CI [0.02–0.26], accounting for 12% of the variance. In other words, maternal education predicts the HLE. The frequency of shared book reading activities at home was not related to number of books for children available at home (ß = 0.04, *p* < 0.72, 95% CI [−0.19–0.28]) to child interest for reading (ß = 0.13, *p* < 0.15, 95% CI [–0.06–0.31]) and to the engagement in reading activities (ß = −0.16, *p* < 0.08, 95% CI [−0.35–0.02]). In other words, maternal education is not related to the child’s characteristics regarding interest and involvement in literacy activities.

None of the environmental factors (e.g., maternal education, number of books, and frequency of shared book reading activities) was significantly related to children’s interest in reading activities and their involvement in reading activities; only age was. In detail, children age was significantly and positively related to their involvement in reading activities ß = 0.10 *p* < 0.01, 95% CI [0.05–0.15] accounting for the 29% of the variance, and negatively related to their interest in reading activities (ß = −0.07 *p* < 0.01, 95% CI [−0.27–−0.02]). Results address RQ1 showing that maternal education accounts for a huge portion of the variance in the frequency of shared book reading activities at home, whereas the number of books for children and children’s characteristics are not. Additionally, only children’s age was significantly related to their interest and engagement in literacy-related activities.

In Model 2a, fitted to answer to the RQ2a, Z-scores for performances at Vocabulary scale Comprehension and Syntax scale Comprehension of TPL (direct assessment) were used. The model, represented in Figure 1, shows a good fit to data (χ2 (10) = 12.49, *p* = 0.253, CFI = 0.983, RMSEA = 0.07, SRMR = 0.07). Both contextual factors (i.e., maternal education, number of children books, and frequency of shared book reading activities) and children characteristics (i.e., interest and engagement in reading activities) were not significantly related to receptive vocabulary and to syntactic comprehension. The two linguistic abilities were linked together (direct effect of receptive vocabulary ß = 0.42 *p* < 0.01, 95% CI [0.22–0.64]). Children’s age was significantly and positively related to both linguistic skills. The magnitude of this relation was higher for vocabulary (ß = 0.14 *p* < 0.01, 95% CI [0.09–0.21]) than for syntactic comprehension (ß = 0.09 *p* < 0.01, 95% CI [0.04–0.15]). Variables included in Model 2a accounted for 45% of the variance in receptive vocabulary and 68% of the variance for syntactic comprehension.

In Model 2b, ran to answer to the RQ2b, fitted values of the Expressive vocabulary Latent variable and Morphosyntactic skill Latent variable that were used. The contribution of children’s age on Expressive vocabulary Latent variable and on morphosyntactic skill Latent variable was controlled. The fit indexes and factor loadings for the Confirmatory Factory Analysis (CFA) for creating Expressive vocabulary and morphosyntactic skill Latent variables used in Model 2b were adequate (FIT indexes for Expressive vocabulary Latent variable: χ2 (5) = 8.10, *p* = 0.15, CFI = 0.989, RMSEA = 0.10, SRMR = 0.02; FIT indexes for morphosyntactic skill Latent variable: CFI = 0.967, RMSEA = 0.12, SRMR = 0.03).

Model 2b showed an acceptable fit (χ2 (5) = 8.91, *p* = 0.11, CFI = 0.98, RMSEA = 0.11, SRMR = 0.05, see Figure 2). Contextual factors, i.e., maternal education, number of books for children at home and frequency of shared book reading activities were not related to vocabulary and to syntactic skills in production. Children engagement in reading activities was significantly related to expressive vocabulary (ß = 0.22 *p* < 0.05, 95% CI [0.01–0.44]) that, in turn, was related to syntactic production (ß = 0.59 *p* < 0.01, 95% CI [0.37–0.82]). Children’s age was significantly and positively related only to Expressive vocabulary Latent variable (ß = 0.12 *p* < 0.01, 95% CI [0.08–0.17]) whereas it was not related to morphosyntactic skill Latent variable (ß = 0.04 *p* < 0.06, 95% CI [−0.003–0.08]). Variables included in Model 2b accounted for the 60% of the variance in Expressive vocabulary Latent variable and 78% of the variance in morphosyntactic skill Latent variable.

## 4. Discussion

This study examined, within the Bioecological theoretical framework [14], the role of both contextual variables and children characteristics on children’s early vocabulary and morphosyntactic skills. According to this framework, personal characteristics, environmental factors, and processes operating between these factors (e.g., quality and quantity of parent-child shared reading interaction) could plausibly contribute to developmental outcomes for children [14]. In detail, in this study, three features of the Home Literacy Environment and three children characteristics were explored: on one hand maternal education, number of books and frequency of shared-book reading, on the other hand, children’s age, interest in reading, and engagement during shared-book reading activities with parents. Although we acknowledge that many other contextual variables and personal characteristics contribute to the language skills of children, the focus of this study has been narrowed to examine very specific factors in a restricted age range (20–36-months).

The current work contributes to the existing literature with three important results. First: maternal education is related to the HLE but not to the children’s characteristics, which in turn resulted independent from HLE. Second, age predicts child interest and engagement. Third, child engagement in literacy activities is related to language skills. The three results are discussed in relation to the existing literature and for their impact on child development and education.

### 4.1. Maternal Education Predicts HLE, but It Is Independent of Child Characteristics and Language Outcomes

Findings of the current study support previous research indicating that family socioeconomic characteristics and in particular maternal education influence children’s literacy environment [22,82]. As expected, maternal education was significantly correlated with all the indicators of the Home Literacy Environment. In fact, our results showed that maternal education correlates both with the availability of educational materials at home and to the frequency of literacy-related activities. Our findings are consistent with those of Yarosz and Bartnett [83] who found that the frequency of shared reading reported by parents was predicted by maternal education, ethnicity, and the number of siblings. Furthermore, we failed to find evidence that maternal education and HLE are related to children’s characteristics, particularly to their interest and engagement in reading and literacy activities. Previous studies examined the relationship of child interest to HLE factors producing somewhat conflicting results [24,74]. Although interactions between parents and children in relation to home literacy and children’s interest can be expected to be closely related, because frequent and stimulating shared-book reading might be assumed to kick-start children’s desire to engage with books independently, we found that children characteristics were not related to contextual aspects. Conversely, our findings are consistent with previous research showing that children’s motivation for reading was not associated with the frequency of storybook reading or library visits [75] and thus is independent of home literacy activities [74]. Intrinsic attitudinal characteristics could indeed reflect individual differences in temperament and/or cognitive ability. As said in other words by Farver, Xu, Eppe, and Lonigan [56], children with high ability might enjoy the challenge of literacy-based activities and thus develop increased motivation to engage in shared-book reading activities. Viewed from a bioecological perspective [14], this pattern of results reflects the importance of personal characteristics, such as children’s interest, alongside environmental factors, in the development of early literacy skills.

The presence of conflicting results in the literature concerning the relation between child interest and engagement in literacy-related activities and HLE could be attributable to different ways in which children’s characteristics are assessed. Most studies used a parent-report measure of child interest, whereas in a small number of recent studies, researchers have used self-report metrics for example by asking children to assign smiley, neutral, or frowning faces to pictures of activities related to literacy, such as storybook reading or writing letters [80]. Hood and colleagues in their study [84], revealed that, while parental reports of children’s interest in reading correlated with both shared reading and parental teaching of letters and words at home, self-report child interest measure did not relate directly to the frequency of shared book reading at home and to any linguistic outcome. Frijters and colleagues [74] suggested that parent reports on children’s literacy interest might be biased or inaccurate and thus produce confounding results. However, in this study, even using a parent-report measure of child interest and engagement in literacy activities, we failed to find relations between HLE measures and child characteristics. This result highlights that child characteristics are not related to contextual aspects. Although a rich and stimulating environment might promote a greater interest and involvement in literacy activities in the child, these seem to be more of intrinsic attitudinal characteristics of a child, emphasizing their importance in a bioecological view. In other words, although we recognize the role of a rich and stimulating HLE, we cannot ignore the individual characteristics of the child that influence his or her engagement and interest during literacy activities. Concerning the relation between maternal education and children’s outcomes, while it is generally accepted that parents’ levels of education have important relationships with children’s academic attainment [85], findings concerning the relationship between mothers’ education and pre-schoolers’ emergent literacy skills have been mixed. Some studies report a strong relationship between maternal education and children’s literacy development [86] while others have found no meaningful difference between children with mothers from either high or low educational backgrounds [87]. The powerful influences of maternal educational level on children’s linguistic outcomes may be a result of the differences in the HLE such as parents’ attitudes and beliefs. This suggests that parental behaviours known to promote literacy development do not necessarily result from educational, social, and economic advantages.

### 4.2. Children’s Age Predicts Children Interest in Literacy Activities and Their Engagement during These Activities

Children’s age resulted to be significantly and positively related to their involvement in shared-book reading whereas it was negatively related with their interest in reading independently, suggesting that as children grow up between 20 and 36 months, they tend to prefer shared-book reading activities over “reading” by themselves.

Additionally, children’s age played, as expected, a significant predictive role for both language measures namely vocabulary and morphosyntactic skill in both comprehension and production highlighting that, words and grammar develop in parallel [88]. Children’s age explained variance ranging from 2 to 34%, in detail: 2% for morphosyntactic comprehension, 34% for vocabulary comprehension, 9% for morphosyntactic production, and 22% for expressive vocabulary. Although these results are not surprising, this is an interesting finding, as all children in this sample were in kindergarten, approximately at the same educational level and the standard deviation of age was less than 6 months. Thus, results show that even a minor variation in months contributed significantly to children’s performance on different measures of early literacy ability, with older children outperforming their younger peers.

### 4.3. Child Engagement Is Related to Language Expressive Skills in Early Stages of Language Development

Our findings failed to show a relationship between child interest for literacy activities and language outcomes in comprehension and production. Neither of the characteristics of the HLE were related to language in this study. On the other hand, the level of child’s engagement significantly predicted expressive vocabulary, which in turn was related to morphosyntactic skills. Even in so young children, active involvement during learning activities is essential for language acquisition. Rather than the presence of educational materials, and the frequency of reading activities, what counts for language acquisition during this highly sensitive stage of development, is the extent to which the children actively participate in learning activities. Development is concerned as activity-dependent and occurs alongside active participation of the child [6]. The new result concludes that, for the first time, this specific relationship is demonstrated for young children and through using multiple measures of language skills. These findings are consistent with previous studies using child-reported literacy interest measures with older children [74] and with studies using different methods of measuring child interest (e.g., parent-report, observation) with preschool children [24,56,73]. The current study extends the findings to very young children and very early stages of language development. The findings are consistent with assertions made by other researchers who have argued that young children’s interests are one set of factors that contribute to early literacy and language development [46,49,73]. We put forward a tentative explanation for the existence of this relationship. For a child who enjoys listening to stories and is involved actively during reading, shared reading episodes might be a good opportunity for learning. This behavior influences the way in which the adult manages these activities. Perhaps, when the child is actively involved, the adults tend to increase the quality and the complexity of the activity, which contribute to teaching a new language but also to further increase the child’s involvement. In fact, the link between children’s engagement and the quality of book reading is almost certainly bidirectional: our speculation is that children more engaged in shared book reading activities could benefit more from the rich input that parents use during this kind of literacy activities and thus obtain greater gains in language development. Observational studies have shown that parents usually tend to use a richer and more complex language in shared reading activities than in other interactions with their children [89], suggesting that more complex linguistic structures to which children are exposed during storybook exposure are important for the development of structural language skills. Shared reading also facilitates learning by providing opportunities for the parent to use questions, expansions, and definitions that focus on language, stories, world knowledge, and emotional reactions [90]. Lastly, the same books can be read over again, thus increasing children’s chances of learning from the books [91]. Similarly, very young children make use of richer vocabulary and continue topic discussions initiated by parents more often during shared book reading than during toy play or mealtimes [92]. Therefore, the extra-textual talk around storybooks contributed by parents and children is a potentially valuable focus for research in understanding how adults scaffold children’s developing language.

## 5. Implications for Practice

The major implication for practice from this study is incorporating children’s interest and involvement into the activities (formal or informal) used as sources of early literacy and language learning opportunities. This includes both the assessment procedures used to identify children’s interests and the types of activities used as interest-based learning opportunities. It is additionally important to conduct parent interventions in language and emergent literacy because parents are engines of change in early intervention programs [93]: parent education is a pathway through which early childhood programs influence child outcomes. It is particularly important to include parents as a source of intervention when attempting to promote children’s language development. Interventions with parents should focus on two aspects: improving features of the HLE such as expanding the range of literacy-related activities, and providing a richer and more stimulating home environment, and improving the child’s engagement during these activities.

A possible way through which parents can foster the child’s engagement in literacy-related activities is through a shift from passive reading to dialogic reading [94], providing interesting materials that take into account the child’s interests. Opportunities to engage in meaningful interactions with interesting and appealing materials and in literacy-related activities are, in fact, critical to successful literacy achievement [47].

Although we can and should attempt to make up for these differences in language input in preschool settings [95], it is difficult for preschool teachers to find time for one-on-one language interactions with their pupils. Clearly, to promote better results, we need to focus our efforts on parents and teachers simultaneously to boost the language development of children.

## 6. Conclusions

The large body of research reviewed, and results of this study provide a relatively convergent picture of the importance of children’s engagement in literacy-related experiences at home for their language development particularly in the first years of life. In the last decades, research has moved beyond attempts to establish that environmental factors are influential in language development, since this is now uncontroversial. Recent studies use a range of measures to clarify what the most significant indicators of environmental influence are and to demonstrate their effects on different components of language and literacy development. The HLE is not a simple construct, but rather includes a range of practices, attitudes, and beliefs, which operate at multiple levels on development, and which are in turn shaped by several characteristics, cognitive and motivational, of the child.

The primary function of socioeconomic status is seen to be in affecting parental attitudes towards literacy-related activities, which in turn predict the types of interactions, which parents engage in with their children and children’s early interest in books and involvement in literacy activities. These HLE practices and child motivational factors may be particularly instrumental in shaping children’s language and emergent literacy skills, the precursors of reading and writing that are strongly predictive of later educational attainment. While other family-level factors, such as resilience to stress, discipline practices, and levels of chaos in the home, may contribute to development in numerous ways, it is only the ‘family as educator’ model that has been causally linked to children’s outcomes [96]. Viewed from a bioecological perspective [14], this pattern of results reflects the importance of both personal characteristics, as children’s engagement in shared-book reading, and environmental characteristics in the development of early language skills.

As said before, the first three years of life represent the most important for brain development, maturation, and specialization [2,6]. As far as we know, this is the first time that contextual aspects and individual child characteristics have been studied in relation to language development at such early ages. To conclude, our results emphasize the role played by both types of factors, contextual and individual, suggesting that these two types of factors should be considered independently in relation to the early stages of children’s language development. Future studies should consider these aspects as separate aspects that concur in determining a child’s developmental trajectory. Several limitations to this work must be considered in interpreting the results. First, replication with a larger sample size is required to confirm our findings. Second, our study design did not allow detecting developmental differences thus, developmental changes should be investigated in future research. Third, since here interest and engagement have been measured through parent-report, in future research child-reported measures should be used to deeply understand their role in language development. Finally, more research is needed on the role of contextual and individual variables in language development especially with children at risk for atypical developmental trajectories. Longitudinal studies might be useful and methodologically appropriate to determine to what extend engagement in reading activities at these early ages would influence the language skills in older ages.

## Figures and Tables

**Figure 1 ijerph-19-00526-f001:**
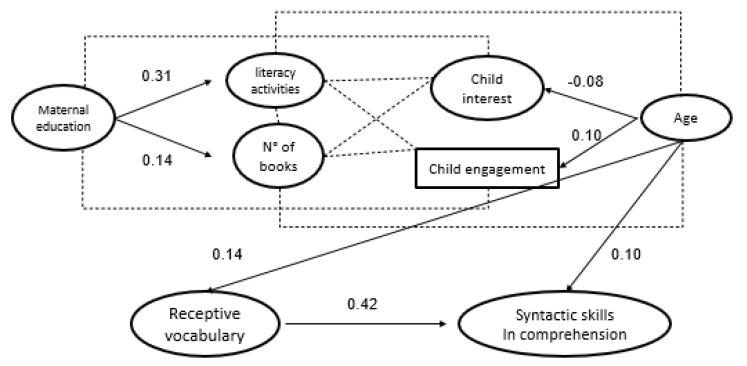
Model 1: contribution of maternal education, shared book reading activities, children’s interest, and engagement in reading activities to vocabulary and syntactic skills in comprehension. Only statistically significant paths are reported.

**Figure 2 ijerph-19-00526-f002:**
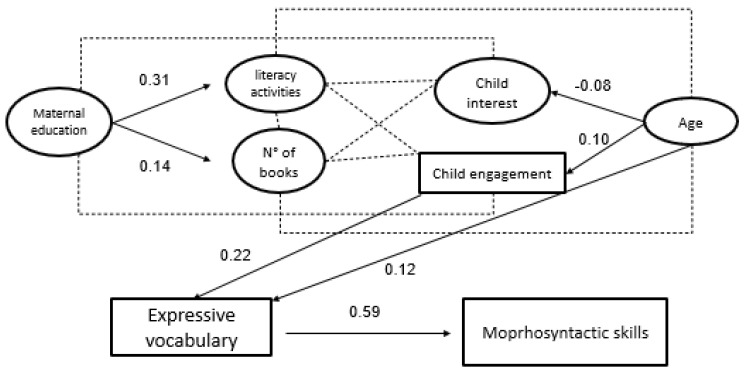
Model 2: contribution of maternal education, shared book reading activities, children’s interest, and engagement in reading activities to vocabulary and syntactic skills in production. Only statistically significant paths are reported.

**Table 1 ijerph-19-00526-t001:** Descriptive statistics.

	Range	Mean (sd)	Skewness	Kurtosis
Age (in months)	20–36	28.2 (5.2)	−0.862	0.061
**TPL (direct assessment)**				
Vocabulary scale Comprehension (range 0–20)	0–19	13.7 (4.7)	−1.863	3.12
Vocabulary scale Production (range 0–20)	7–20	17.2 (2.9)	−1.943	3.91
Syntax scale Comprehension (range 0–20)	0–20	13.1 (6.1)	−0.906	−0.459
Syntax scale Production (range 0–60)	0–52	20.1 (15.6)	0.293	−1.04
**PVB (indirect assessment)**				
Total words (0–650)	23–650	375 (198)	−0.172	−1.43
Total words (percentile)	5–95	57.6 (26.1)	−0.443	−0.886
Mean length of utterance	0–19	6.1 (4.1)	0.906	0.994
Syntax complexity (0–37)	0–37	27.4 (14.3)	−1.06	−0.681
Age of lexical development	16–38	28.4 (6.7)	0.087	−1.30
Lexical Quotient	62–141	100 (16.1)	0.203	0.737

Bold used to highlight the different test (TPL or PVB) indexes.

**Table 2 ijerph-19-00526-t002:** Descriptive statistics of contextual measures.

	Range	Mean (sd)	Skewness	Kurtosis
Maternal education (in years)	8–20	17.11 (2.3)	−1.69	3.92
Number of books for children (range 0–5) *	1–5	2.75 (1.1)	0.625	–0.183
Frequency of shared book reading (range 0–5) **	1–5	4.11 (1.1)	–1.15	0.559

* Note: 0 = none; 1 = between 1 to 20; 2 = between 21 to 40; 3 = between 41 to 60; 4 = between 61 to 80; 5 = more than 80. ** Note: 0 = Never; 1 = Less than once a week; 2 = Once a week; 3 = More than once a week; 4 = Once a day; 5 = More than once a day.

**Table 3 ijerph-19-00526-t003:** Descriptive statistics of children interest in reading activities.

Variable.	Range	Mean (sd)	Skewness	Kurtosis
Engages with books independently (range 0–3) *	0–3	2.36 (0.75)	−1.06	0.917
Child dialogues during reading (range 0–3) *	0–3	2.48 (0.73)	−1.41	1.89
Child interrupts during reading (range 0–3) *	0–3	1.61 (1.1)	−0.254	−0.985
Child invents story during reading (range 0–3) *	0–3	0.82 (0.84)	0.607	−0.638

* Note: 0 = never; 1= sometimes; 2 = often; 3 = always.

**Table 4 ijerph-19-00526-t004:** Zero-Order Correlations between Contextual Variables.

	1.	2.	3.	4.	5.	6.
1. Maternal education	1	0.34 *	0.73 **	−0.07	−0.01	0.16
2. Number of books		1	0.33 *	0.11	0.24	0.18
3. Frequency of shared book reading			1	0.12	−0.14	−0.01
4. Child interest in reading				1	−0.09	−0.39 *
5. Child engagement in reading					1	0.64 **
6. Child age (in months)						1

* *p* < 0.05; ** *p* < 0.01.

## Data Availability

The data that support the findings of this study are available from the corresponding author, R.D., upon reasonable request.

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
