# Peer review of "How Does Toddlers’ Engagement in Literacy Activities Influence Their Language Abilities?"

_ijerph, 2022, doi:10.3390/ijerph19010526_

Round 1
Reviewer 1 Report
The manuscript entitled “How does toddlers’ engagement in literacy activities influence their language abilities?” examined 20 toddlers between 20 and 36 months of age. The authors measured family socioeconomic status (maternal education), the degree of the child’s interest and engagement in literacy activities, and the home literacy environment. The main finding was that the child’s involvement in literacy activities was associated with their language skills (morphosyntax and vocabulary). The authors suggest that caregivers plan activities to fully engage children during reading to boost their language skills. The methods of the study are well-designed, with careful consideration of the results from the prior literature. The findings have important implications for planning early language interventions. While the results of this work are valuable, the paper needs a major revision. Detailed comments are outlined below.
- P. 1, Introduction, paragraph 2: the authors mention theoretical models, but they did not provide specific examples and citations to these models.
- P. 1-7, Introduction: the section is too lengthy. This part of the manuscript should be more focused and significantly shortened.
- P. 16-17, Conclusions: this section is too long and should be shortened substantially. There is information about Neuroconstructivism that did not appear earlier in the text. I suggest moving this section to the Introduction.
- There are numerous repetitions in the text. Most can be eliminated. For example, on p. 13, section 4.1, lines 594-598, it says “[…] our results showed that maternal education correlates both with the availability of educational materials at home and to the frequency of literacy-related activities. In other words, mothers who completed higher-levels of education reported greater numbers of books for children at home and were more likely to involve with their children in literacy-related activities”. The first sentence is clear, and there is no need to repeat it by rephrasing it.
- The manuscript needs to be read and substantially revised by a native speaker of English. I am not a native speaker myself, but I noticed a magnitude of language errors. The errors need to be corrected to comply with the language standards of the IJERPH.
Author Response
Responses to reviewers
We are grateful to the reviewers for their constructive comments. We appreciate the time and effort in reviewing our manuscript. We have tried to adjust the manuscript according to their feedback. Below are our responses to each comment.
Response to Reviewer 1 Comments
Point 1: P. 1, Introduction, paragraph 2: the authors mention theoretical models, but they did not provide specific examples and citations to these models
Response 1: We have explicitly reported some theoretical frameworks with relative references (i.e. Chomsky; Vygotskij; Karmiloff-Smith); see Introduction, paragraph 2.
Point 2: P. 1-7, Introduction: the section is too lengthy. This part of the manuscript should be more focused and significantly shortened.
Response 2: In this version, according to this suggestion, we have shortened the Introduction to improve its readability and fluency. Additionally, as suggested at Point 3, we have moved the section about Neuroconstructivism from Conclusions to the Introduction.
Point 3: P. 16-17, Conclusions: this section is too long and should be shortened substantially. There is information about Neuroconstructivism that did not appear earlier in the text. I suggest moving this section to the Introduction.
Response 3: As suggested, we have moved the section about Neuroconsructivism to the Introduction thus shortening the Conclusions.
Point 4: There are numerous repetitions in the text. Most can be eliminated. For example, on p. 13, section 4.1, lines 594-598, it says “[…] our results showed that maternal education correlates both with the availability of educational materials at home and to the frequency of literacy-related activities. In other words, mothers who completed higher-levels of education reported greater numbers of books for children at home and were more likely to involve with their children in literacy-related activities”. The first sentence is clear, and there is no need to repeat it by rephrasing it.
Response 4: As suggested, we have removed some redundant sentences throughout the manuscript.
Point 5: The manuscript needs to be read and substantially revised by a native speaker of English. I am not a native speaker myself, but I noticed a magnitude of language errors. The errors need to be corrected to comply with the language standards of the IJERPH.
Response 5: If the paper will be accepted, as suggested, we will be keen to send the paper for English revisions to a native speaker to improve its readability. In that way the editing will be made on a more definite proof.
Reviewer 2 Report
Submission title: How does toddlers’ engagement in literacy activities influence their language abilities?
Recommendation: accept after minor revision
Comments: Interesting paper and very well written. Every ideas are very well explained.
- On page 11, the Table 4 should be edit: numbers indicated in the heading of the last to columns should be 5 and 6 in stead of 6 and 7.
- Since this is the first time that contextual aspects and individual child characteristics have been studied in relation to language development at such critical early ages, this reviewer thinks that it would be interesting to do a longitudinal study to determine in what extend engagement in reading activities at these early ages would influence the language skills in older ages. If authors agree, this could be mentioned in the conclusions, when speaking about future studies.
Author Response
We thank reviewer #2 for the nice comments and words of appreciation for our work
Point 1: On page 11, the Table 4 should be edit: numbers indicated in the heading of the last to columns should be 5 and 6 instead of 6 and 7.
Response 1: Done.
Point 2: Since this is the first time that contextual aspects and individual child characteristics have been studied in relation to language development at such critical early ages, this reviewer thinks that it would be interesting to do a longitudinal study to determine in what extend engagement in reading activities at these early ages would influence the language skills in older ages. If authors agree, this could be mentioned in the conclusions, when speaking about future studies.
Response 2: We agree with reviewer #2 about the need of longitudinal studies that could better describe the effects of specific factors, individual and contextual, on early language development, thus we have included this suggestion in the section on future developments.
Round 2
Reviewer 1 Report
The authors have adequately addressed the reviewers’ comments. I recommend this
work for publication. However, an extensive language revision is still required.